# The Infarct-Reducing Effect of the δ_2_ Opioid Receptor Agonist Deltorphin II: The Molecular Mechanism

**DOI:** 10.3390/membranes13010063

**Published:** 2023-01-04

**Authors:** Sergey V. Popov, Alexandr V. Mukhomedzyanov, Leonid N. Maslov, Natalia V. Naryzhnaya, Boris K. Kurbatov, N. Rajendra Prasad, Nirmal Singh, Feng Fu, Viacheslav N. Azev

**Affiliations:** 1Laboratory of Experimental Cardiology, Cardiology Research Institute, Tomsk National Research Medical Center of the Russian Academy of Sciences, 634012 Tomsk, Russia; 2Department of Biochemistry and Biotechnology, Faculty of Science, Annamalai University, Chidambaram 608002, India; 3Department of Pharmaceutical Sciences and Drug Research, Punjabi University, Patiala 147002, India; 4Department of Physiology and Pathophysiology, Fourth Military Medical University, Xi’an 710032, China; 5The Branch of the Institute of Bioorganic Chemistry, Russian Academy of Sciences, 117997 Pushchino, Russia

**Keywords:** heart, ischemia, kinases, K_ATP_ channels, opioid receptors, reperfusion

## Abstract

The search for novel drugs for the treatment of acute myocardial infarction and reperfusion injury of the heart is an urgent aim of modern pharmacology. Opioid peptides could be such potential drugs in this area. However, the molecular mechanism of the infarct-limiting effect of opioids in reperfusion remains unexplored. The objective of this research was to study the signaling mechanisms of the cardioprotective effect of deltorphin II in reperfusion. Rats were subjected to coronary artery occlusion (45 min) and reperfusion (2 h). The ratio of infarct size/area at risk was determined. This study indicated that the cardioprotective effect of deltorphin II in reperfusion is mediated via the activation of peripheral δ_2_ opioid receptor (OR), which is most likely localized in cardiomyocytes. We studied the role of guanylyl cyclase, protein kinase Cδ (PKCδ), phosphatidylinositol-3-kinase (PI3-kinase), extracellular signal-regulated kinase-1/2 (ERK1/2-kinase), ATP-sensitive K^+^-channels (K_ATP_ channels), mitochondrial permeability transition pore (MPTP), NO synthase (NOS), protein kinase A (PKA), Janus 2 kinase, AMP-activated protein kinase (AMPK), the large conductance calcium-activated potassium channel (BK_Ca_-channel), reactive oxygen species (ROS) in the cardioprotective effect of deltorphin II. The infarct-reducing effect of deltorphin II appeared to be mediated via the activation of PKCδ, PI3-kinase, ERK1/2-kinase, sarcolemmal K_ATP_ channel opening, and MPTP closing.

## 1. Introduction

The incidence of acute myocardial infarction and mortality did not decrease in the last decade, despite the increasing use of modern interventional and conservative methods of treatment [1]. Percutaneous coronary intervention (PCI) provides >95% restoration of blood flow in the infarct-related coronary artery, but some patients still die [2]. The use of thrombolysis and PCI in clinical practice has made reperfusion injury of the heart come to the fore, which causes the death of about 50% of cardiomyocytes in the ischemia/reperfusion (I/R) zone [3]. Currently, there are no drugs in clinical practice capable of effectively preventing the reperfusion damage to the heart. Therefore, the search for novel drugs capable of effectively preventing cardiac reperfusion injury is considered a high priority of modern pharmacology and experimental cardiology.

The results of our preliminary studies indicated that OR agonists can prevent not only ischemic, but also reperfusion injury of the heart [4]. There are potential reasons to believe that OR agonists can serve as a prototype for creating a new class of drugs that increase cardiac resistance to reperfusion. In 2009, we found that the selective δ_2_ OR agonist deltorphin II could prevent cardiac I/R injury when administered before ischemia [5]. In 2021, we found that deltorphin II can prevent reperfusion injury of the heart [4]. Our studies indicated that the infarct-reducing effect of deltorphin II is mediated via the activation of δ_2_ OR in cardiomyocytes [4]. However, the molecular mechanism of the cardioprotective effect of deltorphin II in reperfusion remains unknown.

The objective of this research was to study the intracellular signaling mechanism(s) of the cardioprotective effect of deltorphin II in reperfusion.

## 2. Materials and Methods

### 2.1. Materials

Male Wistar rats were purchased from the Institute of Cytology and Genetics SB RAS (Novosibirsk) and used in the present study. The animals were housed in 12:12 h light/dark conditions at the temperature of 24 ± 2 °C. The rats received commercial laboratory feed pellets, along with drinking water ad libitum. All procedures were governed by the Directive 2010/63/EU of the European Parliament and the Guide for the Care and Use of Laboratory Animals published by the US National Institutes of Health (NIH Publication No. 85-23, revised 1996). The Ethical Committee of the Cardiology Research Institute of the Tomsk National Research Medical Center granted the approval for this study (Protocol N° 207 dated 23 December 2020).

### 2.2. Coronary Artery Occlusion and Reperfusion

The rats were anesthetized with chloralose (50 mg/kg, intraperitoneally, Sigma-Aldrich, St. Louis, MO, USA) and connected to a SAR-830 Small Animal Ventilator artificial respirator (CWE, Inc., Ardmore, PA, USA). They underwent thoracotomy, and the pericardium was removed. The ligature of the left descending coronary artery was superimposed 1–2 mm below the left atrial appendage according to a previously published method [6]. Coronary occlusion was verified by ST segment elevation. The right carotid artery was cannulated for measuring blood pressure, which was registered using an SS13L pressure transducer (Biopac System Inc., Goleta, CA, USA) coupled with an MP35 electrophysiological device (Biopac System Inc., Goleta, CA, USA). The same apparatus was used to record the ECG. Quantitative data processing was performed using INSTBSL-W software of Biopac System Inc., (Goleta, CA, USA).

After 45 min of ischemia, the silk ligature was loosened, and the restoration of the blood flow was confirmed by the appearance of epicardial hyperemia. The duration of reperfusion was 2 h.

### 2.3. Detection of Infarct Size and the Area at Risk

The identification of the necrosis zone and the risk zone was carried out according to the method proposed by Neckar et al. [7]. After reperfusion, the hearts were removed and rinsed with a syringe through the cannulated aorta with a saline solution containing 125 IU/mL of heparin. To determine the area at risk, the ligature was re-tightened, and the myocardium was stained through the aorta with 5% potassium permanganate; after washing, 1 mm thick heart sections perpendicular to the longitudinal axis were prepared. The necrosis zone was isolated from the risk zone by staining with a 1% solution of 2,3,5-triphenyl tetrazolium chloride for 30 min at 37 °C. After staining, the slices were placed in a 10% formaldehyde solution for 1 day. The day after staining, the right ventricle was removed, and left ventricular sections were scanned on both sides with an HP Scanjet G2710 scanner. The area at risk and the infarction zone were determined by a computerized planimetric method using the ImageJ software (Wayne Rasband, Research Services Branch of the National Institute of Mental Health, NIH). The infarct size is expressed as a percentage of the size of the AAR, i.e., as the ratio infarct size/area at risk.

### 2.4. Pharmacological Agents

Opioid receptor antagonists and inhibitors were administered intravenously 10 min prior to reperfusion. The selective δ_2_ OR agonist deltorphin II was administered intravenously 5 min prior to reperfusion at a dose of 0.12 mg/kg [5]. The choice of the doses of OR antagonists and the administration schedule were based on our previously published data [4,5]. Naltrexone, which is an antagonist of all types of OR, was used at a dose of 5 mg/kg [5]. The non-selective OR antagonist naloxone methiodide, which does not penetrate the blood–brain barrier, was injected at a dose of 5 mg/kg [5]. The selective peptide δ OR antagonist TIPP [ψ] was administered at a dose of 1 mg/kg [5]. The selective δ_2_ OR antagonist naltriben mesylate was used at a dose of 0.3 mg/kg [5].

We used inhibitors of kinases and other test substances which are involved in the regulation of cardiac tolerance to I/R. L-NAME, an inhibitor of NO-synthase (NOS), was injected at a dose of 10 mg/kg [8]. Chelerythrine was used at a dose of 5 mg/kg to inhibit PKC [5]. Rottlerin, a PKCδ inhibitor, was injected at a dose of 0.3 mg/kg [9]. The ERK1/2-kinase inhibitor PD-098059 was injected at a dose of 0.5 mg/kg [10]. The Janus 2 kinase inhibitor AG490 was used at a dose of 3 mg/kg [11]. Wortmannin, a PI3-kinase inhibitor, was administered at a dose of 0.025 mg/kg [12]. The AMP-activated protein kinase (AMPK) inhibitor compound C was injected at a dose of 0.25 mg/kg [13]. The NO-sensitive soluble guanylyl cyclase inhibitor ODQ was injected at a dose of 1 mg/kg [14]. The PKA inhibitor H-89 was administered at a dose of 10 µg/kg [15]. Glibenclamide was used to block the K_ATP_ channel at a dose of 1 mg/kg [16]. The mitochondrial K_ATP_ channel inhibitor 5-hydroxydecanoate (5-HD) was administered at a dose of 5 mg/kg, and the sarcolemmal K_ATP_ channel inhibitor HMR 1098 was injected at a dose of 3 mg/kg [17,18]. The BK_Ca_ channel inhibitor paxilline was administered at a dose of 1.5 mg/kg [19]. The MPTP opener atractyloside was administered at a dose of 5 mg/kg [15]. The preferential ^•^OH scavenger 2-mercaptopropionyl glycine (MPG) was injected at a dose of 20 mg/kg [20]. The superoxide radical scavenger Tempol was used at a dose of 30 mg/kg [21].

All used inhibitors and blockers were administered intravenously 10 min before reperfusion.

The peptides deltorphin II and TIPP[ψ] were provided free by PolyPeptide Laboratories (San Diego, CA). HMR 1098 was synthesized and provided by Sanofi-Aventis Deutschland GmbH (Frankfurt, Germany). Chloralose, 2,3,5-triphenyl tetrazolium chloride, L-NAME, MPG, naltrexone, naloxone methiodide, rottlerin, glibenclamide, 5-HD were purchased from Sigma-Aldrich (St. Louis, MO, USA). Tempol, compound C, ODQ, H-89, paxilline, hydroxypropyl-β-cyclodextrin were purchased from Tocris (Bristol, UK). PD-098059, chelerythrine, AG 490, wortmannin were synthesized and provided free by LCLabs Company (Woburn, MA, USA). The ELISA kits (CEA003Ge cAMP, CEA577Ge cGMP) were purchased from Cloud-Clone Corporation (Wuhan, China). We purchased atractyloside from MedChemExpress (Princeton, NJ, USA). The water-insoluble compounds were dissolved in 0.1 mL of DMSO and then diluted in 0.9 mL of 20% hydroxypropyl-β-cyclodextrin.

### 2.5. Experimental Protocol

The experiments performed in 528 rats which were divided into 44 groups: 1—control (45 min of ischemia and 2 h reperfusion); 2—naltrexone; 3—naloxone methiodide; 4—TIPP[ψ]; 5—naltriben mesylate; 5—deltorphin II; 6—deltorphin II + naltrexone; 7—deltorphin II + naloxone methiodide; 8—deltorphin II + TIPP[ψ]; 9—deltorphin II + naltriben mesylate; 10—chelerythrine; 11—rottlerin; 12—wortmannin; 13—PD98059; 14—H-89; 15—AG490; 16—L-NAME; 17—compound C; 18—ODQ; 19—glibenclamide; 20—5-HD; 21—HMR1098; 22—atractyloside; 23—paxilline; 24—MPG; 25—tempol; 26—deltorphin II + chelerythrine; 27—deltorphin II + rottlerin; 28—deltorphin II + wortmannin; 29—deltorphin II + PD98059; 30—deltorphin II + H-89; 31—deltorphin II + AG490; 32—deltorphin II + L-NAME; 33—deltorphin II + Compound C; 34—deltorphin II + ODQ; 35—deltorphin II + glibenclamide; 36—deltorphin II + 5-HD; 37—deltorphin II + HMR1098; 38—deltorphin II + atractyloside; 39—deltorphin II + paxilline; 40—deltorphin II + MPG; 41—deltorphin II + tempol; 42—control (cAMP and cGMP level); 43—I/R (cAMP and cGMP level); 44—deltorphin II (cAMP and cGMP level). Each experimental group included 12 rats.

### 2.6. Detection of Cyclic Nucleotides

The myocardial tissue was frozen in liquid nitrogen. The extraction of cyclic nucleotides was carried out as we previously described [22]. The cAMP and cGMP levels were determined in in the area at risk 15 min after the onset of reperfusion with deltorphin II and without deltorphin II and also in the left ventricle of rats without coronary artery occlusion. The cAMP and cGMP levels were determined in myocardial homogenates using standard commercial ELISA kits (CEA003Ge cAMP, CEA577Ge cGMP) from Cloud-Clone Corporation (Wuhan, China).

### 2.7. Statistical Analysis

The results are expressed as mean ± standard error of the mean (SEM) from the indicated number of experiments. Data analysis was performed using STATISTICA 13. The homogeneity of the variances was assessed by the Levene’s test. To identify statistically significant differences between independent groups and a single control, we used one-way ANOVA followed by Dunnett’s post hoc test. To identify statistically significant differences in dependent groups, we used one-way repeated-measures ANOVA followed by Dunnett’s post hoc test. To identify statistically significant differences in independent groups, we used one-way ANOVA followed by Bonferroni post hoc test. All the statistical hypotheses were accepted when the significance level was less than *p* < 0.05.

## 3. Results

It was demonstrated that deltorphin II and opioid receptor antagonists had no effect on the hemodynamics (Table 1). Most inhibitors also had no effect on the hemodynamics in doses utilized (Table 1). The Janus 2 kinase inhibitor AG490 decreased the heart rate (Table 1). The NO synthase inhibitor L-NAME decreased the heart rate and increased the systolic blood pressure (Table 1).

None of the used OR antagonists had an effect on the infarct size/area at risk ratio (Figure 1).

The selective δ_2_ opioid receptor agonist deltorphin II decreased the infarct size by 37% (Figure 2).

Pretreatment with the non-selective OR antagonist naltrexone and naloxone methiodide which does not cross the blood–brain barrier (BBB) promoted the removal of the infarct-reducing effect of deltorphin II (Figure 2). The selective δ OR antagonist TIPP (ψ) and the selective δ_2_ OR antagonist naltriben also contributed to the removal of the infarct-reducing effect of deltorphin II (Figure 2).

Pretreatment with the PKC inhibitor chelerythrine or the selective PKCδ inhibitor rottlerin promoted the elimination of the infarct-sparing effect of deltorphin II (Figure 3).

Pretreatment with the PI3-kinase inhibitor wortmannin promoted the disappearance of the infarct-reducing effect of deltorphin II (Figure 3). The ERK1/2-kinase inhibitor PD98059 contributed to the removal of the infarct-limiting effect of deltorphin II (Figure 3). The PKA inhibitor H-89 did not eliminate the cardioprotective effect of deltorphin II (Figure 3). The Janus 2 kinase inhibitor AG490 also did not eliminate the infarct-limiting effect of deltorphin II (Figure 3). The NOS inhibitor L-NAME and the AMPK inhibitor compound C did not reverse the infarct-reducing effect of deltorphin II (Figure 3). The NO-sensitive soluble guanylyl cyclase inhibitor ODQ promoted the disappearance of the cardioprotective effect of deltorphin II (Figure 3).

The non-selective K_ATP_ channel blocker glibenclamide contributed to the removal of the infarct-sparing effect of deltorphin II (Figure 4).

The mitochondrial K_ATP_ channel blocker 5-HD did not reverse the infarct-reducing effect of deltorphin II (Figure 4). In contrast, the sarcolemmal K_ATP_ channel blocker HMR 1098 contributed to the removal of the infarct-limiting effect of deltorphin II (Figure 4). The MPTP opener atractyloside also promoted the elimination of the cardioprotective effect of deltorphin II (Figure 4). The BK_Ca_ channel inhibitor paxilline did not abolish the infarct-reducing effect of deltorphin II (Figure 4).

We studied the role of ROS in deltorphin II-induced cardioprotection. It was determined that the preferential ^•^OH scavenger MPG did not reverse the infarct-reducing effect of deltorphin II (Figure 4). The superoxide scavenger Tempol did not abolish the cardioprotective effect of deltorphin II (Figure 4).

It should be noted that all inhibitors had no effect on the infarct size/area at risk ratio (Figure 5 and Figure 6).

Reperfusion caused a decrease in the cAMP level and an increase in the cGMP level in the area at risk (Table 2). Deltorphin II promoted a five-fold increase in the cGMP level in the area at risk compared to the control group of animals with I/R of the heart.

## 4. Discussion

It was shown that the infarct-reducing effect of deltorphin II is mediated via the activation of peripheral δ_2_ OR. These data are consistent with our previously published data [4]. This δ_2_ OR appears to be localized on the sarcolemma of cardiomyocytes, because deltorphin II increases the resistance of isolated cardiomyocytes to anoxia/reoxygenation via the stimulation of δ_2_ OR [4].

It was found that PKCδ is involved in the infarct-sparing effect of deltorphin II. This result is consistent with our previously published data indicating that the cardioprotective effect of opioids was associated with protein kinase activation in I/R of the heart [4]. Consequently, PKCδ is involved in the cardioprotective effect of opioids during reperfusion.

It was also demonstrated that the infarct-reducing effect of deltorphin II is associated with the activation of PI3-kinase and ERK1/2-kinase. The obtained result is consistent with previously published data of the Gross’ group showing that PI3-kinase and ERK1/2-kinase participate in the cardioprotective effect of the κ OR agonist U-50488 at reperfusion [23].

There is evidence that PKA, AMPK, and Janus 2 are involved in morphine-induced preconditioning [11,15,24]. However, we demonstrated that AMP-activated protein kinase, Janus2, and PKA did not participate in the infarct-reducing effect of deltorphin II. Consequently, it could be proposed that the molecular mechanism of the cardioprotective effect of morphine differs from that of the cardioprotective effect of deltorphin II. Morphine, unlike deltorphin II, readily crosses the blood–brain barrier, thereby the cardioprotective effect of morphine could be mediated via stimulation of central opioid receptors. Therefore, the molecular mechanism of the cardioprotective effect of morphine could be different from the mechanism of deltorphin-triggered cardioprotection. Morphine was used before ischemia [11,15,24], thereby we cannot rule out the possibility that the molecular mechanism of opioid-induced preconditioning could be different from the mechanism of opioid-induced postconditioning.

It was previously reported that NOS is involved in the cardioprotective effect of the opioid Eribis peptide 94 [25]. However, our data indicate that NOS is not involved in the cardioprotective effect of deltorphin II.

It is generally accepted that the mitochondrial K_ATP_ channel is involved in the infarct-limiting effect of opioids when used before cardiac I/R [5,26] and at reperfusion [23]. However, the results of our study indicate that the sarcolemmal K_ATP_ channel can also participate in the infarct-limiting effect of deltorphin II in reperfusion. It was previously reported that MPTP and the BK_Ca_ channel are involved in the cardioprotective effect of the κ OR agonist U-50488 [27]. We also obtained evidence that the infarct-reducing effect of deltorphin II is mediated via MPTP closing. The BK_Ca_ channel did not participate in the cardioprotective effect of deltorphin II.

Reactive oxygen species could be intracellular signaling molecules [28]. There is evidence that ROS are involved in the cardioprotective effect of the δ OR agonist SNC-121 in reperfusion [20]. However, our studies demonstrated that the infarct-sparing effect of deltorphin II in reperfusion does not involve ROS. Consequently, the signaling mechanism of the cardioprotective effect of deltorphin II could be different from the signaling mechanism of the protective effect of SNC-121. It is possible that the cardioprotective effect of SNC-121 is the result of central OR stimulation and the cardioprotective effect of deltorphin II is the result of peripheral δ_2_ OR activation; therefore, the molecular mechanisms of their effects could be different.

We previously reported that the non-selective peptide δ OR agonist dalargin can increase the cGMP level in the area at risk during ischemia [22]. Deltorphin II had a similar effect at reperfusion. It has been reported that there are two main source of cGMP formation in cardiomyocyte: natriuretic peptide receptor and soluble NO-sensitive guanylyl cyclase [29]. Pretreatment with the soluble NO-sensitive guanylyl cyclase inhibitor ODQ completely removed the cardioprotective effect of deltorphin II. This was an unexpected result, since NOS inhibition did not affect the infarct-sparing effect of deltorphin II. It should be noted that soluble guanylyl cyclase can be activated not only by NO but also by CO. It is possible that deltorphin II activated heme oxygenase-1 and increased CO production and CO stimulation of guanylyl cyclase, leading to cardioprotection. There is also indirect evidence that ODQ can inhibit natriuretic peptide-sensitive guanylyl cyclase [30].

It was found that deltorphin II contributed to a reduction in the myocardial cAMP level in reperfusion (Table 2). This result is in agreement with previous findings that the peptide δ OR agonist dalargin reduced the myocardial cAMP level in ischemia [22]. There are data that opioid receptors are coupled with G_i/o_ proteins [31]. The stimulation of G_i/o_ proteins promotes the inhibition of the adenylyl cycle [31]. It could be proposed that deltorphin II promoted the inhibition of the adenylyl cycle via the activation of G_i/o_ proteins.

## 5. Conclusions

Our study showed that the cardioprotective effect of deltorphin II in reperfusion is mediated via the activation of peripheral δ_2_ OR, which is probably localized in cardiomyocytes. The infarct-reducing effect of deltorphin II is mediated via the activation of PKCδ, PI3-kinase, ERK1/2-kinase, sarcolemmal K_ATP_ channel opening, MPTP closing. PKA, Janus 2, AMP-activated protein kinase, NOS, ROS, the BK_Ca_ channel, and the mitochondrial K_ATP_ channel are not involved in the infarct-limiting effect of deltorphin II in reperfusion. We also demonstrated that guanylyl cyclase plays an important role in the cardioprotective effect of deltorphin II in reperfusion.

## Figures and Tables

**Figure 1 membranes-13-00063-f001:**
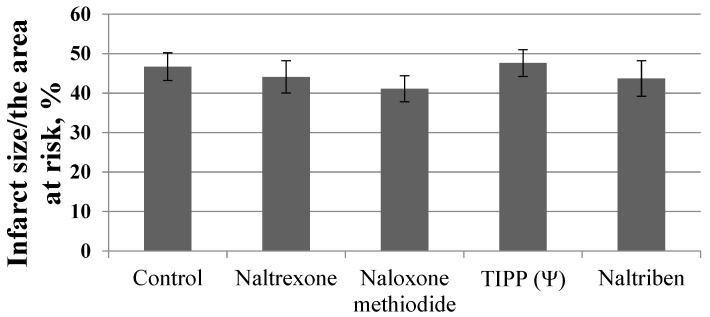
Effect of opioid receptor antagonists on the infarct size/area at risk ratio (*Y*-axis). Data represent the means ± SEM. Each study group included 12 rats.

**Figure 2 membranes-13-00063-f002:**
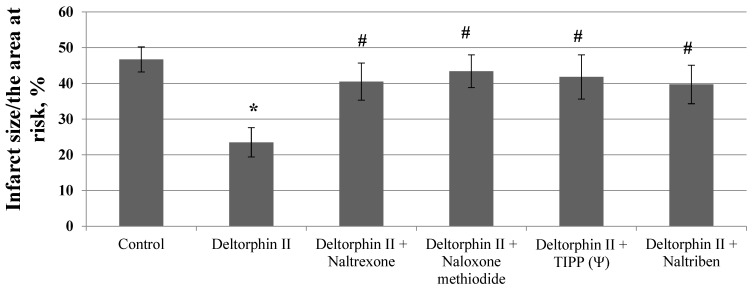
Involvement of opioid receptors in the mechanism of the infarct-limiting effect of deltorphin II (*Y*-axis). Data represent the means ± SEM. * *p* < 0.05 vs. the control group; # *p* < 0.05 vs. the deltorphin II group. Each study group included 12 rats.

**Figure 3 membranes-13-00063-f003:**
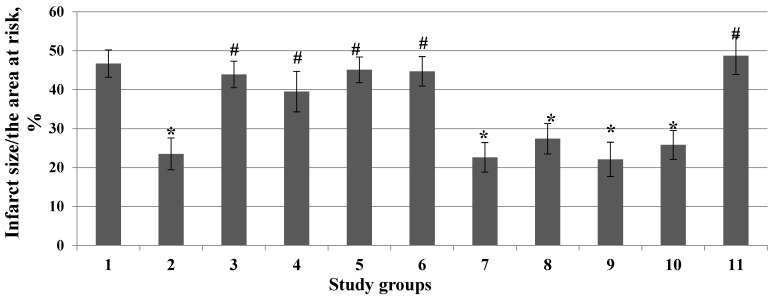
Involvement of kinases and NOS in the infarct-limiting effect of deltorphin II (*Y*-axis). 1—Control; 2—Deltorphin II; 3—Deltorphin II + Chelerythrine; 4—Deltorphin II + Rottlerin; 5—Deltorphin II + Wortmannin; 6—Deltorphin II + PD98059; 7—Deltorphin II + H-89; 8—Deltorphin II + AG490; 9—Deltorphin II + L-NAME; 10—Deltorphin II + Compound C; 11—Deltorphin II + ODQ. Data represent the means ± SEM. * *p* < 0.05 vs. the control group; # *p* < 0.05 vs. the deltorphin II group. Each study group included 12 rats.

**Figure 4 membranes-13-00063-f004:**
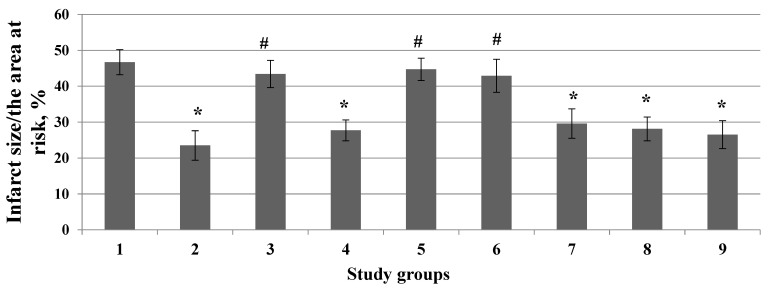
The involvement of K_ATP_ channels, MPT pore, the BK_Ca_ channel, and ROS in the infarct-reducing effect of deltorphin II (*Y*-axis). 1—Control; 2—Deltorphin II; 3—Deltorphin II + Glibenclamide; 4—Deltorphin II + 5-HD; 5—Deltorphin II + HMR1098; 6—Deltorphin II + Atractyloside; 7—Deltorphin II + Paxilline; 8—Deltorphin II + MPG; 9—Deltorphin II + Tempol. Data represent the means ± SEM. * *p* < 0.05 vs. the control group; # *p* < 0.05 vs. the deltorphin II group. Each study group included 12 rats.

**Figure 5 membranes-13-00063-f005:**
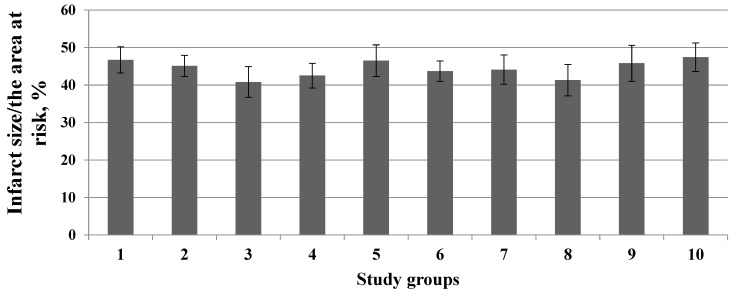
Effect of inhibitors of kinases and molecular structures on the infarct size/area at risk ratio (*Y*-axis). 1—Control; 2—Chelerythrine; 3—Rottlerin; 4—Wortmannin; 5—PD98059; 6—H-89; 7—AG490; 8—L-NAME; 9—Compound C; 10—ODQ. Data represent the means ± SEM. Each study group included 12 rats.

**Figure 6 membranes-13-00063-f006:**
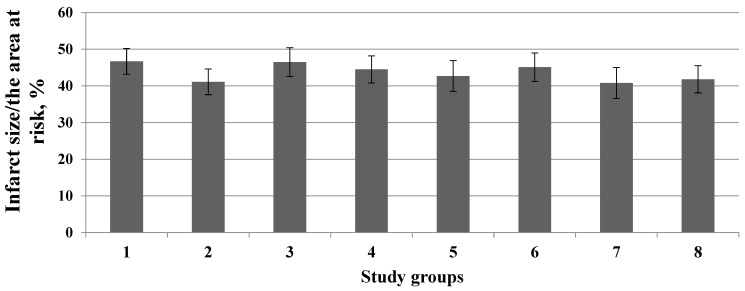
Effect inhibitors of kinases and molecular structures on the infarct size/area at risk ratio (*Y*-axis). 1—Control; 2—Glibenclamide; 3—5-HD; 4—HMR1098; 5—Atractyloside; 6—Paxilline; 7—MPG; 8—Tempol. Data represent the means ± SEM. Each study group included 12 rats.

**Table 1 membranes-13-00063-t001:** Hemodynamics data.

	Before Ischemia	Before Reperfusion	After 30 min Reperfusion	After 2 h Reperfusion
Heart rate (beats/min)				
Control	365 ± 4	359 ± 4	352 ± 5	342 ± 9
Deltorphin II	367 ± 5	362 ± 5	354 ± 4	348 ± 8
Naltrexone	359 ± 8	353 ± 6	348 ± 5	340 ± 7
Naloxone methiodide	355 ± 6	349 ± 4	345 ± 5	336 ± 6
TIPP [ψ]	366 ± 3	358 ± 4	353 ± 4	344 ± 7
Naltriben	358 ± 5	352 ± 4	346 ± 3	337 ± 7
L-NAME	366 ± 5	335 ± 5 *	318 ± 4 *	304 ± 9 *
Chelerythrine	361 ± 4	354 ± 3	347 ± 5	339 ± 6
Rottlerin	370 ± 3	363 ± 5	356 ± 4	345 ± 7
PD098059	362 ± 4	355 ± 4	350 ± 5	341 ± 8
AG490	367 ± 3	340 ± 4 *	337 ± 5 *	321 ± 7 *
Wortmannin	360 ± 5	354 ± 3	349 ± 4	339 ± 9
ODQ	365 ± 4	359 ± 4	354 ± 5	345 ± 7
H-89	367 ± 5	361 ± 5	356 ± 4	344 ± 8
Compound C	364 ± 4	360 ± 6	354 ± 5	346 ± 9
Glibenclamide	359 ± 4	354 ± 3	349 ± 5	338 ± 6
5-HD	368 ± 3	363 ± 3	358 ± 4	347 ± 5
HMR1098	362 ± 4	356 ± 5	350 ± 4	336 ± 7
Paxilline	357 ± 5	352 ± 4	345 ± 5	335 ± 8
Atractyloside	366 ± 4	360 ± 4	356 ± 3	345 ± 5
MPG	361 ± 3	355 ± 4	347 ± 4	337 ± 7
Tempol	356 ± 5	351 ± 4	344 ± 3	334 ± 8
Mean systolic blood pressure (mmHg)				
Control	126 ± 5	122 ± 3	119 ± 4	114 ± 6
Deltorphin II	122 ± 4	119 ± 4	115 ± 5	109 ± 7
Naltrexone	125 ± 3	120 ± 5	117 ± 3	111 ± 5
Naloxone methiodide	128 ± 4	124 ± 4	120 ± 5	113 ± 7
TIPP[ψ]	124 ± 3	119 ± 5	116 ± 3	111 ± 8
Naltriben	120 ± 3	117 ± 4	113 ± 3	106 ± 5
L-NAME	123 ± 4	145 ± 3 *	155 ± 5 *	161 ± 6 *
Chelerythrine	126 ± 3	123 ± 4	118 ± 4	113 ± 5
Rottlerin	125 ± 4	121 ± 4	116 ± 5	111 ± 6
PD098059	128 ± 3	124 ± 5	120 ± 3	112 ± 7
AG490	127 ± 4	140 ± 3 *	138 ± 4 *	134 ± 5 *
Wortmannin	126 ± 5	121 ± 4	117 ± 3	113 ± 7
ODQ	122 ± 3	117 ± 3	114 ± 5	108 ± 5
H-89	127 ± 4	123 ± 4	118 ± 5	113 ± 8
Compound C	124 ± 6	121 ± 4	119 ± 3	114 ± 5
Glibenclamide	128 ± 5	125 ± 5	121 ± 7	117 ± 6
5-HD	121 ± 3	118 ± 4	114 ± 6	109 ± 5
HMR1098	125 ± 3	120 ± 5	117 ± 3	112 ± 7
Paxilline	119 ± 6	115 ± 4	111 ± 5	105 ± 8
Atractyloside	128 ± 3	123 ± 4	118 ± 4	113 ± 6
MPG	124 ± 4	121 ± 3	117 ± 4	110 ± 7
Tempol	121 ± 5	116 ± 4	112 ± 3	107 ± 8

Data represent the means ± SEM. * *p* < 0.05 vs. the control group.

**Table 2 membranes-13-00063-t002:** Effect of deltorphin II on the cAMP and cGMP levels in ischemia/reperfusion of the heart.

Group	cAMP (nmol/g)	cGMP (nmol/g)
Control	18.2 ± 1.1	9 ± 1.1
I/R	13.1 ± 3.9 *	24 ± 4.4 *
Deltorphin II	11.8 ± 2.8 ^#^	44 ± 4.7 *^,#^

Data represent the means ± SEM. * *p* < 0.05 vs. the control group; ^#^
*p* < 0.05 vs. the I/R group; I/R, ischemia/reperfusion.

## Data Availability

The data presented in this study are available on request from the corresponding author.

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
