# Peer review of "The Infarct-Reducing Effect of the δ2 Opioid Receptor Agonist Deltorphin II: The Molecular Mechanism"

_membranes, 2023, doi:10.3390/membranes13010063_

Round 1

Reviewer 1 Report

The study aims at the importance of Deltorphin II as an important drug to reduce infarct size in rats via Delta opioid receptors and important signaling molecules. The study is well designed and experiments are nicely explained. The results clearly report the effect of Deltorphin II. I have some comments and suggestions however:

1. Deltorphin II is reported to be a selective agonist for DOR but why did the authors not applied SNC 80 which is another stronger and more potent antagost for DOR. It has been shown that Deltorphin II can activate  other signaling receptors like somatostatin receptors. In another experiment if the similar reduction in infarct size be seen using SNC 80 that would truly conclude the effect to be mediated via DOR.

2. Minor edit Line 34: Did not "decrease" instead of "decreased"

Author Response

Dear colleague, thank you very much for reviewing our manuscript. We tried to take into account your recommendations and recommendations of Reviewer 2 as much as possible.

The study aims at the importance of Deltorphin II as an important drug to reduce infarct size in rats via Delta opioid receptors and important signaling molecules. The study is well designed and experiments are nicely explained. The results clearly report the effect of Deltorphin II. I have some comments and suggestions however:

  1. Deltorphin II is reported to be a selective agonist for DOR but why did the authors not applied SNC 80 which is another stronger and more potent antagost for DOR. It has been shown that Deltorphin II can activate other signaling receptors like somatostatin receptors. In another experiment if the similar reduction in infarct size be seen using SNC 80 that would truly conclude the effect to be mediated via DOR.

We used deltorphin II because it is the selective δ2-opioid receptor agonist. There is no evidence that SNC 80 is the selective δ2-opioid receptor agonist. In addition, it was reported that peptide opioid receptor agonist in small doses cannot penetrate the blood barrier:

Polonskii V.M., Yarygin K.N., Krovosheev O.G., Moskovkin G.N., Vinogradov V.A. Effect of the antiulcerative action (central or peripheral) of the synthetic enkaphalin analog dalargin in experimental cysteamine-induced duodenal ulcer in rats. Bull. Exp. Biol. Med. 1987; 103(4): 488-490.

Maslov L.N., Lishmanov Yu.B. Permeability of the blood-brain barrier for opioid peptides. Exp. Clin. Pharmacol. 2017; 80(6): 39-44. In Russian

We proposed that deltorphin II at a dose of 0.12 mg/kg can activate peripheral opioid receptors. We would like to activate only peripheral opioid receptors. The non-peptide δ-opioid receptor agonist SNC 80 probably stimulates both peripheral and central opioid receptors [PMID: 15936000]. We found that deltorphin II at a dose of 0.12 mg/kg activate peripheral opioid receptors. We demonstrated that the cardioprotective effect of deltorphin II is mediated via activation of δ2-opioid receptor. Somatostatin receptors are probably not involved in the cardioprotective effect of deltorphin II.

  1. Minor edit Line 34: Did not "decrease" instead of "decreased"

We corrected. The Reviewer 2 requested several changes to the text. We have highlighted them in red.

Sincerely yours, Leonid N. Maslov

Reviewer 2 Report

The paper entitled The Infarct-Reducing Effect of the δ2 Opioid Receptor Agonist 2 Deltorphin II: the Molecular Mechanism is interesting and provides evidence of the drug's mechanism of action, however, I have some comments:
I think it is important to change the name of section 2.2, I also suggest separating paragraph two of section 2.2 to a section with a specific name.
Additionally, it would be necessary to add a paragraph that mentions the experimental design where the groups to be used and the total number of rodents used are described.
I suggest merging the information from sections 2.3 and 2.5.,
It is necessary to mention the route of administration of the pharmacological agents described in point 2.3
In statistical analysis, the authors mention that they used an ANOVA, it is understood that it is one way, however for table 1, this would not be the appropriate analysis,
I consider that it is necessary to mention in point 2.6 the analysis that was used for the results shown in Table 2
the authors should explain more clearly the information in figures 1 and 2 and mention the importance of both figures, I think that the information in figure 2 should appear before figure 1. Same comment for the information in figures 5 and 6
the authors should describe the results in a better way, not say directly “eliminates”, “did not eliminate” etc…, Authors could give a preamble of the meaning of the axes in the figures

in table 2, because cAMP data are lower for deltorphin II than for I/R, and for cGMP, because it increases with deltorphin II with respect to I/R
explain in the discussion,
 in discussion:
The authors mention in line 236 “Consequently, the molecular mechanism of the cardioprotective effect of morphine differs from the cardioprotective effect of deltorphin II”, not only should they mention that the results are different from other works, the authors must give a reasonable argument of because they are different. The same commentary for the information in the following paragraphs of the discussion.
 Minor comments:
Line 36, there is a point before reference 1.

In the information on line 154, add space with the data from the previous table

Author Response

Dear colleague, thank you very much for reviewing our manuscript. We tried to take into account your recommendations and recommendations of Reviewer 1 as much as possible.

The paper entitled The Infarct-Reducing Effect of the δ2 Opioid Receptor Agonist 2 Deltorphin II: the Molecular Mechanism is interesting and provides evidence of the drug's mechanism of action, however, I have some comments:

I think it is important to change the name of section 2.2, I also suggest separating paragraph two of section 2.2 to a section with a specific name.

We separated Section 2.2. into two parts.

Additionally, it would be necessary to add a paragraph that mentions the experimental design where the groups to be used and the total number of rodents used are described.

I suggest merging the information from sections 2.3 and 2.5.

We prepared section 2.5. experimental protocol. We merged sections 2.3 and 2.5.

It is necessary to mention the route of administration of the pharmacological agents described in point 2.3

We added this information.

In statistical analysis, the authors mention that they used an ANOVA, it is understood that it is one way, however for table 1, this would not be the appropriate analysis,

We corrected statistical analysis.

I consider that it is necessary to mention in point 2.6 the analysis that was used for the results shown in Table 2

We corrected statistical analysis.

the authors should explain more clearly the information in figures 1 and 2 and mention the importance of both figures, I think that the information in figure 2 should appear before figure 1.

We added more information in Results and Discussion. We corrected Figure legends.

Same comment for the information in figures 5 and 6 the authors should describe the results in a better way, not say directly “eliminates”, “did not eliminate” etc…, Authors could give a preamble of the meaning of the axes in the figures

We tried as much as possible to remove “eliminated”, “abolished”, “reversed”

In table 2, because cAMP data are lower for deltorphin II than for I/R, and for cGMP, because it increases with deltorphin II with respect to I/R explain in the discussion,

We discussed this issue in Discussion.

In discussion:

The authors mention in line 236 “Consequently, the molecular mechanism of the cardioprotective effect of morphine differs from the cardioprotective effect of deltorphin II”, not only should they mention that the results are different from other works, the authors must give a reasonable argument of because they are different. The same commentary for the information in the following paragraphs of the discussion.

We altered this sentence. Of course, we did not repeated studies performed by Gross E.R. et al. (2006), Li L. et al, (2011), Dorsch, M. et al, (2016). However, we have no reason to believe that these researchers performed their research incorrectly. In addition, morphine, unlike deltorphin II, readily crosses the blood brain barrier thereby the cardioprotective effect of morphine could be mediated via stimulation of central opioid receptors. Therefore, the molecular mechanism of the cardioprotective effect of morphine could be different from the mechanism of the deltorphin-triggered cardioprotection. Morphine was used before ischemia thereby we cannot rule out the possibility that the molecular mechanism of opioid-induced preconditioning could be different from the mechanism of opioid-induced postconditioning.

Minor comments:

Line 36, there is a point before reference 1.

We removed this point.

In the information on line 154, add space with the data from the previous table

We added space.

Sincerely yours, Leonid N. Maslov

Round 2

Reviewer 1 Report

I accept the manuscript in its present form